# Prediction of the Uric Acid Component in Nephrolithiasis Using Simple Clinical Information about Metabolic Disorder and Obesity: A Machine Learning-Based Model

**DOI:** 10.3390/nu14091829

**Published:** 2022-04-27

**Authors:** Hao-Wei Chen, Yu-Chen Chen, Jung-Ting Lee, Frances M. Yang, Chung-Yao Kao, Yii-Her Chou, Ting-Yin Chu, Yung-Shun Juan, Wen-Jeng Wu

**Affiliations:** 1Graduate Institute of Clinical Medicine, College of Medicine, Kaohsiung Medical University, 100, Shih-Chuan 1st Road, Kaohsiung 80708, Taiwan; chanhoward21@hotmail.com (H.-W.C.); jennis7995@hotmail.com (Y.-C.C.); yihech@kmu.edu.tw (Y.-H.C.); juanuro@gmail.com (Y.-S.J.); 2Department of Urology, Kaohsiung Municipal Ta-Tung Hospital, Kaohsiung, 80145, Taiwan; 3Department of Urology, Kaohsiung Medical University Hospital, Kaohsiung Medical University, Kaohsiung 80708, Taiwan; 4Si Wan College, National Sun-Yat Sen University, Kaohsiung 80424, Taiwan; celeste@g-mail.nsysu.edu.tw; 5School of Nursing, University of Kansas, Kansas City, KS 66160, USA; fyang3@kumc.edu; 6Department of Electrical Engineering, National Sun-Yat Sen University, Kaohsiung 80424, Taiwan; cykao@mail.ee.nsysu.edu.tw; 7Department of Business Management, National Sun Yat-Sen University, Kaohsiung 80424, Taiwan; chu880102@gmail.com

**Keywords:** uric acid, nephrolithiasis, machine learning, gout, diabetes mellitus, glomerular filtration rate, obesity

## Abstract

There is a great need for a diagnostic tool using simple clinical information collected from patients to diagnose uric acid (UA) stones in nephrolithiasis. We built a predictive model making use of machine learning (ML) methodologies entering simple parameters easily obtained at the initial clinical visit. Socio-demographic, health, and clinical data from two cohorts (A and B), both diagnosed with nephrolithiasis, one between 2012 and 2016 and the other between June and December 2020, were collected before nephrolithiasis treatment. A ML-based model for predicting UA stones in nephrolithiasis was developed using eight simple parameters—sex, age, gout, diabetes mellitus, body mass index, estimated glomerular filtration rate, bacteriuria, and urine pH. Data from Cohort A were used for model training and validation (ratio 3:2), while data from Cohort B were used only for validation. One hundred and forty-six (13.3%) out of 1098 patients in Cohort A and 3 (4.23%) out of 71 patients in Cohort B had pure UA stones. For Cohort A, our model achieved a validation AUC (area under ROC curve) of 0.842, with 0.8475 sensitivity and 0.748 specificity. For Cohort B, our model achieved 0.936 AUC, with 1.0 sensitivity, and 0.912 specificity. This ML-based model provides a convenient and reliable method for diagnosing urolithiasis. Using only eight readily available clinical parameters, including information about metabolic disorder and obesity, it distinguished pure uric acid stones from other stones before treatment.

## 1. Introduction

Urolithiasis is a common disease worldwide. The prevalence and incidence of these urinary stones are increasing [1]. This rise in global prevalence is a major contributing factor to increases in healthcare costs associated with nephrolithiasis [2]. Moreover, recurrence rates range between 50% and 80%, depending on the type of stone [3].

In urolithiasis, uric acid (UA) stones account for 10% to 15% of all stones [4]. Unlike other types of stones, uric acid stones have been associated with several features of metabolic syndrome and nutrient partitioning disorders including diabetes mellitus and obesity [5,6,7,8,9]. In addition, unlike other stones which require surgery, most uric acid stones can be treated conservatively [10,11]. Thus, it is important to be able to tell the difference between uric acid and non-uric acid stone diseases prior to surgery. Uric acid stones are usually diagnosed through the analysis of stones after they have been retrieved via surgery or medical expulsive therapy. Although recent studies have reported that dual-energy computed tomography (CT) can be an accurate test for distinguishing uric acid from non-uric acid stone diseases prior to treatment [12], these studies have used different cutoffs and CT machine types [13,14], making replication difficult. In addition, dual-energy CT produces a relatively high radiation dose and the equipment is prohibitively expensive for most clinics around the world.

Previously, we used structural equation modeling to construct a model for the pathway analysis of UA nephrolithiasis identifying several simple clinical factors affecting UA stone disease [6]. Among these clinical factors, we found eight variables, namely sex, age, gout, diabetes mellitus (DM), body mass index (BMI), estimated glomerular filtration rate (eGFR), bacteriuria, and urine pH, that have indirect and direct effects on the formation of uric acid stones [6,7]. The aim of this study was to build a reliable diagnostic model making use of machine learning using these variables including information about metabolic disorder and obesity to predict the uric acid component in nephrolithiasis and test it. The eight variables chosen were simple clinical factors easily obtained during the patient’s first clinical visit.

## 2. Materials and Methods

### 2.1. Selection and Description of Participants

In this retrospective study, we collected data for two cohorts (A and B), patients diagnosed with nephrolithiasis at Kaohsiung Medical University Hospital, Kaohsiung, Taiwan between January 2012 and December 2016 and between June 2020 and December 2020, respectively. Cohort A was a group of patients we enrolled in a previous study [6,7]. The participant selection criteria for Cohort B were the same as those for Cohort A. We included patients diagnosed with nephrolithiasis who had received genitourinary surgery (ureteroscopic lithotripsy, percutaneous nephrolithotomy, or open nephrolithotomy). We excluded any of these patients if they had genitourinary tract tumors, kidney transplants, genitourinary tract anomalies, recurrent nephrolithiasis, mixed stones containing more than one stone component, or renal replacement therapy, such as hemodialysis. Patients younger than 18 years and those without detailed medical records were also excluded.

This study was approved by the Kaohsiung Medical University Chung-Ho Memorial Hospital Institutional Review Board (KMUHIRB-E(I)-20210061) and was conducted according to the principles of the Declaration of Helsinki. The informed consent requirement was waived by Kaohsiung Medical University Chung-Ho Memorial Hospital Institutional Review Board (KMUHIRB-E(I)-20210061) due to the retrospective nature of this study and the minimal risk involved.

### 2.2. Data Description and Processing

In addition to the response variable, eight variables were used for data analysis and prediction model building. These included two sociodemographic variables (age and sex), three health-related variables (BMI and the comorbidities DM and gout), and three clinical variables (urine pH and bacteriuria, and eGFR). Socio-demographic, health, and clinical information was collected prior to treatment for nephrolithiasis.

■Response variable

Stones were analyzed using infrared spectroscopy performed by a medical technologist and confirmed by two urologists. They were defined as 1 (uric acid stone) or 0 (other stone).

■Socio-demographic characteristics

Age and gender were recorded. Age was entered as a numeric variable and gender as a binary variable, 1 (male) or 0 (female).

■Health information

Health information items were patient BMIs, and whether or not the patient had a history of DM or gout. BMI was entered as a numeric variable with one decimal fraction. DM was entered as 1 (having a history) or 0 (having no history). Gout was recorded similarly, 1 (having a history) or 0 (having no history).

■Clinical information

Urinalysis results, including bacteriuria and urine pH, were entered. Urine pH was entered as a numeric variable with one decimal fraction, while bacteriuria was entered as 1 (having bacteriuria) or 0 (having no bacteriuria). eGFR was calculated using the isotope dilution mass spectrometry traceable Modification of Diet in Renal Disease formula, where (eGFR [mL/min/1.73 m^2^] = 175 × (Scr) ^−1.154^ × (Age) ^−0.203^ × [0.742 if female]) [15].

### 2.3. Model Development, Fitting, and Evaluation

A model for predicting uric acid stones in nephrolithiasis was developed using machine learning methodologies [16]. The mathematical representation of this model can be expressed as follows:$$y=\frac{1}{1+e^{f\left(x\right)}}$$
where x represents the input vector of the eight decision variables, and y the model output, which takes a value between 0 and 1. The model output is expressed as the likelihood that stones were composed of uric acid. The decision variables $x_{1},\ldots ,\;x_{8}$ carry the values of “gender”, “age”, “eGFR”, “urine pH”, “BMI”, “DM”, “gout” and “bacteriuria”, respectively. The function $f\left(x\right)$ has the following form:$$f\left(x\right)=a_{1}f_{1}\left(x_{1}\right)+\cdots +a_{5}f_{5}\left(x_{5}\right)+a_{6}f_{6}\left(x_{6},x_{7},x_{8}\right)+b,$$
where $a_{1},\;\cdots ,a_{6},\;b$ are model parameters, and each $f_{i}\left(\cdot \right)$ function is a nonlinear function generated by a fully connected two-layer neural network. Each neural network had an output dimension of 20 for the first layer and 1 for the second layer. More specifically, functions *f*_1_ to *f*_6_ have the following forms:$$f_{i}\left(x_{i}\right)=\sigma_{i,2}\left(A_{i,2}\sigma_{i,1}\left(A_{i,1}x_{i}\right)+B_{i,2}\right),\;i=1,\ldots ,\;5$$
$$f_{6}\left(x_{6},\;x_{7},x_{8}\right)=\sigma_{6,2}\left(A_{6,2}\sigma_{6,1}\left(A_{6,1}x_{6}+A_{7,1}x_{7}+A_{8,1}x_{8}\right)+B_{6,2}\right),$$
where each of $A_{1,1}$ to $A_{8,1}$ is a 20-by-1 vector variable which contains 20 parameters, each of $A_{1,2}$ to $A_{6,2}$ is a 1-by-20 vector variable which also contains 20 parameters, and each of $B_{1,2}$ to $B_{6,2}$ is a scalar parameter; $\sigma_{i,1}$ and $\sigma_{i,2}$, i=1,…, 6 are standard nonlinear activation functions commonly used in building a neural network. In total, the nonlinear function $f\left(x\right)$ contains 293 trainable model parameters. The model was built using the Python programming language and its built-in libraries commonly used for machine learning practice. The variables were trained by applying the built-in optimization engine with a weighted binary cross entropy function as its objective.

Cohort A data were used for model training and model validation, while Cohort B data were only used for validation. Group A data were randomly divided into training (60%) and validation (40%) sets. We calculated the prevalence of patients with pure uric acid and non-uric acid stone in both training and validation datasets. There was very little statistical difference between training and validation sets with regard to sociodemographic characteristics, medical history, and bacteriuria.

The model parameters were trained based on the training dataset (Cohort A) and predictive power was evaluated using the validation datasets (Cohorts A and B), using an area under the curve (AUC) in the receiver operating characteristic (ROC) analysis. The optimal cutoff point on the ROC curve was determined based on Youden’s index [17], which in turn was used to calculate the sensitivity, specificity, positive predictive value, and negative predictive value for the prediction models applied to the validation data. Finally, the prediction model was applied to data obtained from the Cohort B to determine whether the uric-acid stone patients in Cohort B could be correctly identified by the model.

## 3. Results

### 3.1. Characteristics of Study Samples

Cohort A consisted of 1098 patients, 146 (13.3%) with pure uric acid stones and 952 (86.7%) with non-uric acid stones (Table 1 and Table 2). Cohort B consisted of 71 patients, including three male patients with pure uric acid stones (Table 3 and Table 4). Statistical similarity between patients with non-uric acid nephrolithiasis in Cohort A and those in B was observed. It is also observed that the mean values of the chosen variables (or the “percentages” in the case of binary variables) are statistically different between patients with pure uric acid stones and those with non-uric acid stones. Student’s *t*-test was applied to verify how significant these differences are, and the resulting *p*-values are also reported in Table 1, Table 2, Table 3 and Table 4. All patients in Cohort A and Cohort B did not take certain medications, such as potassium citrate, sodium citrate, sodium bicarbonate, potassium acid phosphate, and acetazolamide, which could impact urine pH one year prior to the study data collection.

### 3.2. Model Performance

The model we developed exhibited good discriminatory power in predicting who would have pure uric acid nephrolithiasis, which it achieved using only a small amount of information that can be or is often collected at the initial patient clinical visit. The model achieved a validation AUC of 0.842 (95% CI, 0.800–0.885) when applied to the Cohort A validation dataset. The optimal cut-off value, which was determined by Youden’s index for pure uric acid nephrolithiasis, was 0.470. Using this cut-off value, the model had a sensitivity of 0.848 (95% CI, 0.756–0.939) and a specificity of 0.748 (95% CI, 0.704–0.792). The positive predictive value (PPV) was 0.343 (95% CI, 0.266–0.419) and the negative predictive value (NPV) was 0.969 (95% CI, 0.950–0.989). The overall correction ratio was 76.13%.

The model was also applied to patients in Cohort B. The AUC of the ROC curve was found to be 0.936 (95% CI, 0.854–1.0). The optimal cut-off value for this ROC curve was 0.688. Using this cut-off value, the model had a sensitivity of 1.0 (95% CI, 1.0–1.0) and a specificity of 0.912 (95% CI, 0.844–0.979). The PPV was 0.333 (95% CI, 0.025–0.641) and the NPV was 1.0 (95% CI, 1.0–1.0). The overall correction ratio was 91.55%. Even when a cut-off value of 0.470 obtained from Cohort A was applied, the model was able to correctly predict all three pure uric acid nephrolithiasis patients in Cohort B, achieving a 1.0 sensitivity, though the specificity was reduced to 0.647 and the overall correction ratio reduced to 66.2%.

Note that the validation set from the cohort group A has 13.41% uric acid stone patients and 86.59% non-uric acid stone patients. Therefore, applying a “random guess” predictive mechanism (i.e., those with 50% sensitivity and 50% specificity) on this set of patients would result in a PPV of 0.1341 and a NPV of 0.8659. Likewise, for patients in cohort B, the PPV and NPV would be 0.0423 and 0.9577, respectively. Our model achieved a PPV of 0.343 and an NPV of 0.969 for the validation set from the cohort group A, and a PPV of 0.333 and an NPV of 1.0 for the cohort group B. Although there is still room for improvement, these results clearly demonstrate the predictive power of our model.

The validation ROC curves of the model are shown in Figure 1.

## 4. Discussion

In urolithiasis disease, uric acid stones account for 10% to 15% of all urinary stones. Because the chemical properties of uric acid stones are very different from those of other stones (particularly calcium-containing stones), the treatment and prevention of uric acid stones could be improved with a diagnostic approach that can exclude non-uric acid stones [10,11]. This study showed that eight simple clinical variables can be used to build a diagnostic model to distinguish pure uric acid stones from non-uric acid ones with very good accuracy. These variables, including age, sex, past medical history, routine urine analysis, and eGFR, can be easily obtained during the first visit to an emergency room or clinic. These findings can help first-line medical staff to distinguish uric acid stones from non-uric acid stones when patients are diagnosed as having nephrolithiasis. This knowledge could make possible the timely oral administration of chemolysis agents to dissolve the uric acid stones.

A meta-analysis of studies investigating the prediction of uric acid stones using dual-energy CT recently showed that this approach had a highly accurate prediction rate [12]. However, the studies reviewed used different methods and machines. For example, in an early retrospective study on CT image prediction for uric acid stones, Ascenti et al. used 120 kV dual-energy CT and post-processing software (Syngo Kidney Stone, Siemens Healthcare) to predict uric acid stones [18]. Zhang et al. performed a prospective study in 2016 using dual-source dual-energy computed tomography to predict the stone components [19]. There are some problems with the use of CT machines. First, the CT machines used in previous studies were different, with some not commonly used in clinical practice. Second, radiation exposure should be kept to a minimum. In low-dose non-contrast abdominal CT for ureter stone diagnosis, some studies limited the radiation dose to be no more than 3 mSv, while others have limited it to only 1–1.5 mSv [20]. A relatively high radiation dose (8–10 mSv) was used by Zhang et al. to analyze stone components [19]. Third, dual-energy CT machines are more expensive than single-energy CT machines. Most abdominal CT scans are single-energy. Although single-energy CT scans were used to predict uric acid stones in 2021, many sophisticated post-CT image processing methods, such as Laplacian filtering, were required [21]. Thus, image-based diagnostic tools may be impractical in clinical use. We believe our model and methodology, which use only simple clinical variables, is easier to perform and avoids subjecting patients to radiation. The model can also be easily applied in clinics or even in areas where health professionals and medical resources, including CT machines, are not widely available.

It is generally believed that having a large number of decision variables is necessary to successfully build a predictive model based on machine learning algorithms. However, having many decision variables often makes it difficult to determine possible causality [22]. In a recent study, Kazemi et al. used more than 40 variables to build diagnostic tools based on machine learning to predict the stone components [23]. Among these variables, some were not confirmed to be related to the formation of stones, such as marital status and smoking. In a previous study, we used a structural equation model to analyze the relationship between uric acid stones and variables that had some evidence for urolithiasis formation [6,7]. We found eight variables including information about metabolic disorder and obesity to be related to the development of uric acid nephrolithiasis. In this study, we used these eight variables to build a diagnostic model based on machine learning methodologies. This study found that our model had good predictive power and worked well. Compared with previous methods [23], the model we use here requires fewer computational resources and can be used after the initial clinical visit of the patient to provide a rapid and reliable alternative diagnostic tool for the identification of pure uric acid nephrolithiasis.

Although the AUCs of our diagnostic model for the two groups of patients were relatively high, there were still some false-positive outcomes. Some studies have proposed that uric acid crystals form the nidus for the deposition of other stone components [24,25,26]. Additionally, one epidemiological study found patients with hyperuricosuria and gouty diathesis to have a higher incidence of non-uric acid nephrolithiasis [27]. These findings might explain the false positive results. Our model also predicted uric acid stones in some patients who were finally diagnosed with non-uric acid stones. We found that these patients had similar etiologies and clinical characteristics to patients with uric acid nephrolithiasis, so even when misdiagnosed by our model, these patients would more than likely benefit from the same treatment and nephrolithiasis prevention measures.

This study has some limitations. First, the urine pH in this study may have some bias. It is known that the urine pH may be affected by some medications and dietary habits, which were not explicitly considered in the study. This is a subject to be investigated in future studies. Moreover, as we have pointed out in the previous section, the lower prevalence of pure uric stones naturally leads to the tendency of low PPV and high NPV. Our model more than doubled the PPV from a random guess, and yet only reached around 34%. There is room for further improvement, and this will be a focus of future studies. Finally, this study did not assess the efficiency of chemolysis medical treatment in patients who were predicted to have uric acid stones and we did not assess the stone recurrence. In the future, a prospective study can be conducted to evaluate the efficiency of chemolysis treatment on patients predicted to have uric acid stones by our current model. More advanced models based on machine learning methodologies might be further developed to predict uric acid stone recurrence.

## 5. Conclusions

In summary, this ML-based model provides a simple, convenient and reliable method for the diagnosis of uric acid stones. With only eight easily obtained clinical parameters including information about metabolic disorder and obesity, this ML-based model can distinguish pure uric acid stones from other stones without advanced equipment before urolithiasis treatment. This diagnostic approach may help to optimize timely treatment strategies for urolithiasis.

## Figures and Tables

**Figure 1 nutrients-14-01829-f001:**
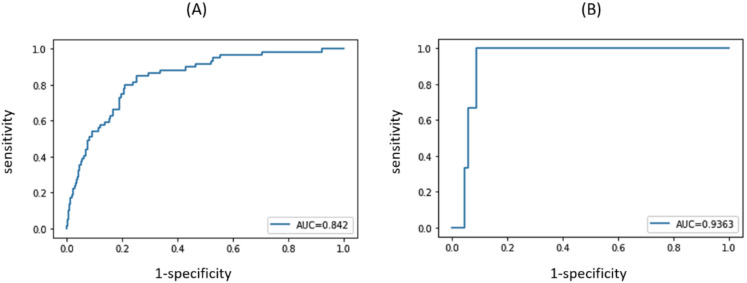
Validation receiver operating characteristic (ROC) curves of the proposed machine learning (ML)-based model on Cohort A and B patients. (**A**): ROC curve for Cohort A. (**B**): ROC curve for Cohort B.

**Table 1 nutrients-14-01829-t001:** Descriptive analysis of the Cohort A included in model training and validation sets.

	Training Set (60%)				Validation Set (40%)			
	Pure Uric Acid (*n* = 87, 13.22%)	Non-Uric Acid (*n* = 571, 86.78%)	Total Patients (*n* = 658)	*p*-Value	Pure Uric Acid (*n* = 59, 13.41%)	Non-Uric Acid (*n* = 381, 86.59%)	Total Patients (*n* = 440)	*p*-Value
Gender				0.095				0.258
male	66 (75.86%)	382 (66.90%)	448 (68.09%)		44 (74.58%)	256 (67.19%)	300 (68.18%)	
female	21 (24.14%)	189 (33.10%)	210 (31.91%)		15 (25.42%)	125 (32.81%)	140 (31.82%)	
Age				<0.001				<0.001
≤45	8 (9.20%)	184 (32.22%)	192 (29.18%)		8 (13.56%)	99 (25.98%)	107 (24.32%)	
45~65	49 (56.32%)	308 (53.94%)	357 (54.25%)		32 (54.24%)	223 (58.53%)	255 (57.95%)	
>65	30 (34.48%)	79 (13.84%)	109 (16.57%)		19 (32.20%)	59 (15.49%)	78 (17.73%)	
DM				<0.001				0.003
with	26 (29.89%)	74 (12.96%)	100 (15.20%)		16 (27.12%)	48 (12.60%)	64 (14.55%)	
without	61 (70.11%)	497 (87.04%)	558 (84.80%)		43 (72.88%)	333 (87.40%)	376 (85.45%)	
Gout				0.046				<0.001
with	5 (5.75%)	12 (2.10%)	17 (2.58%)		9 (15.25%)	9 (2.36%)	18 (4.09%)	
without	82 (94.25%)	559 (97.90%)	641 (97.42%)		50 (84.75%)	372 (97.64%)	422 (95.91%)	
Bacteriuria				0.033				0.155
with	7 (8.05%)	97 (16.99%)	104 (15.81%)		6 (10.17%)	67 (17.59%)	73 (16.59%)	
without	80 (91.95%)	474 (83.01%)	554 (84.19 %)		53 (89.83%)	314 (82.41%)	367 (83.41%)	

Abbreviations: DM, diabetes mellitus.

**Table 2 nutrients-14-01829-t002:** Age, BMI, urine pH, and eGFR of all study samples in Cohort A.

	Pure Uric Acid (15.76%) Mean (SD) 95% CI	Non-Uric Acid (84.24%) Mean (SD) 95% CI	*p*-Value
Age	60.44 (12.52) (58.41–62.47)	52.75 (12.69) (51.94–53.55)	<0.001
BMI	25.63 (3.80) (25.02–26.25)	25.38 (3.53) (25.16–25.61)	0.4297
Urine pH	5.51 (0.54) (5.43–5.60)	6.09 (0.77) (6.04–6.14)	<0.001
eGFR	55.13 (29.45) (50.35–59.90)	80.14 (29.37) (78.27–82.01)	<0.001

Abbreviations: BMI, body mass index; eGFR, estimated glomerular filtration rate; SD, standard deviation; CI, confidence interval.

**Table 3 nutrients-14-01829-t003:** Patient characteristics in Cohort B.

	Pure Uric Acid (*n* = 3, 4.23%)	Non-Uric Acid (*n* = 68, 95.77%)	Total Patients (*n* = 71)	*p*-Value
Gender				0.170
male	3 (100.00%)	41 (60.29%)	44 (61.97%)	
female	0 (0.00%)	27 (39.71%)	27 (38.03%)	
Age				0.812
≤45	1 (33.33%)	15 (22.06%)	16 (22.54%)	
45~65	1 (33.33%)	37 (54.41%)	38 (53.52%)	
>65	1 (33.33%)	16 (23.53%)	17 (23.94%)	
DM				0.095
with	2 (66.67%)	16 (23.53%)	18 (25.35%)	
without	1 (33.33%)	52 (76.47%)	53 (74.65%)	
Gout				0.034
with	1 (33.33%)	3 (4.41%)	4 (5.63%)	
without	2 (66.67%)	65 (95.59%)	67 (94.37%)	
Bacteriuria				0.632
with	0 (0.00%)	5 (7.35%)	5 (7.04%)	
without	3 (100.00%)	63 (92.65%)	66 (92.96%)	

Abbreviations: DM, diabetes mellitus.

**Table 4 nutrients-14-01829-t004:** Age, BMI, urine pH, and eGFR of all study samples in Cohort B.

	Pure Uric Acid (4.23%) Mean (SD), 95% CI	Non-Uric Acid (95.77%) Mean (SD) 95% CI	*p*-Value
Age	56.67 (14.01) (40.72–72.52)	54.72 (13.78) (51.44–57.99)	0.812
BMI	28.77 (6.11) (21.86–35.68)	26.80 (4.75) (25.67–27.93)	0.489
Urine pH	5.0 (0) (5.0–5.0)	6.24 (0.88) (6.03–6.44)	0.019
eGFR	65.20 (18.55) (44.21–86.19)	76.91 (32.07) (69.28–84.53)	0.534

Abbreviations: BMI, body mass index; eGFR, estimated glomerular filtration rate; SD, standard deviation; CI, confidence interval.

## Data Availability

The datasets generated and/or analyzed during the current study are available from the corresponding author upon reasonable request.

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
