# Peer review of "Prediction of the Uric Acid Component in Nephrolithiasis Using Simple Clinical Information about Metabolic Disorder and Obesity: A Machine Learning-Based Model"

_nutrients, 2022, doi:10.3390/nu14091829_

Round 1

Reviewer 1 Report

This is a machine learning-based approach to improve the predictability of uric acid stone. This approach is unique and addresses some gaps in the field of diagnostics and therapeutics. However, there are limitations to this approach, let alone this study. To this end, I would like the authors to expand on their approach and provide a rationale for selecting their decision variables.

Please address and expand on the following:

  • Urine pH levels can be significantly impacted by certain medications and dietary habits. How does the model overcome these challenges? Acute exposure versus chronic exposure?
  • Bacteriuria can also change urine pH. Were there any differences in the urine pH in the training dataset versus the validation dataset, recognizing that bacteriuria was significantly different
  • Bacteriuria can be caused by a urinary tract infection. Why not use UTI diagnosis over bacteriuria?
  • All your decision variables are consistent with an increased risk of uric acid reabsorption and an increased risk of hyperuricemia. Why not use SUA levels as one of the decision variables?
  • Please comment on the sample size of patients with uric acid stone and how it may have impacted the PPV and NPV
  • Please specify the statical test used to compare between groups in all your tables.

Reviewer 2 Report

Well written presentation of an observational, retrospective study (9 pages, 4 tables, 1 figure, 27 references) assessing the predictive value of a formula using 8 clinically relevant and accessible variables: age, gender, BMI, DM, eGFR, gout, urinary pH, bacteriuria (model training group of 658, and validation group of 440 patients with analyzed kidney stones) to identify UA stones.

Development and calculation of the predictive formula should be explained in more detail.

The references are quoted in a inconsistent way: please follow the ICMJE guidelines or the Journal's citation suggestions.

Round 2

Reviewer 1 Report

This is a much-improved version. The authors addressed nearly all the comments. I still have some minor comments. 

1- Please make sure you spell out pH correctly. It was spelled Ph in some sections. 

2- Please acknowledge how low or high prevalence of an outcome can impact the PPV and NPV in your limitations section. 
